# Radiotherapy in Follicular Lymphoma Staged by ^18^F-FDG-PET/CT: A German Monocenter Study

**DOI:** 10.3390/biomedicines9050561

**Published:** 2021-05-17

**Authors:** Imke E. Karsten, Gabriele Reinartz, Michaela Pixberg, Kai Kröger, Michael Oertel, Birte Friedrichs, Georg Lenz, Hans Theodor Eich

**Affiliations:** 1Department of Radiation Oncology, University Hospital of Muenster, 48149 Muenster, Germany; gabriele.reinartz@ukmuenster.de (G.R.); Kai.Kroeger@ukmuenster.de (K.K.); Michael.Oertel@ukmuenster.de (M.O.); hans.eich@ukmuenster.de (H.T.E.); 2Department of Medicine A, Hematology, Oncology and Pneumology, University Hospital of Muenster, 48149 Muenster, Germany; birte.friedrichs@ukmuenster.de (B.F.); georg.lenz@ukmuenster.de (G.L.); 3Department of Nuclear Medicine, University Hospital of Muenster, 48149 Muenster, Germany; michaela.pixberg@ukmuenster.de

**Keywords:** follicular lymphoma, non-Hodgkin lymphoma, radiation therapy, chemotherapy, ^18^F-FDG-PET/CT, oncological imaging, therapeutic evaluation

## Abstract

This retrospective study examined the role of ^18^F-fluorodeoxyglucose-positron emission tomography/computed tomography (^18^F-FDG-PET/CT) in stage-related therapy of follicular lymphomas (FL). Twelve patients each in stages I and II, 13 in stage III and 11 in stage IV were treated in the Department of Radiation Oncology, University Hospital of Muenster, Germany from 2004 to 2016. Radiotherapy (RT), as well as additional chemoimmunotherapy were analyzed with a median follow-up of 87.6 months. Ultrasound (US), CT and ^18^F-FDG-PET/CT were used to determine progression-free survival (PFS), overall survival (OS) and lymphoma-specific survival (LSS) over 5- and 10- years. 23 of 24 patients with stage I/II (95.8%) had complete remissions (CR) and 17 of 24 patients with stages III/IV FL showed CR (70.8%). 5- and 10-year PFS in stages I/II was 90.0%/78.1% vs. 44.3%/28.5% in stages III/IV. 5- and 10-year OS rates in stages I/II was 100%/93.3% vs. 53.7%/48.4% in stages III/IV. 5- and 10-year LSS of stages I/II was 100%/93.8% vs. 69.2%/62.3% in stages III/IV. FL of stages I/II, staged by ^18^F-FDG-PET/CT, revealed better survival rates and lower risk of recurrence compared to studies without PET/CT-staging. Especially, patients with PET/CT proven stage I disease showed significantly better survival and lower relapses rates after RT.

## 1. Introduction

Follicular lymphoma (FL) is the most common subtype of indolent Non-Hodgkin lymphoma (NHL) and, with 20 to 25%, the second most common subtype of NHL in the western industrialized countries [1] and in the USA. In accordance, the most recent epidemiological study from the USA describes an incidence rate of 3.2 new cases per 100,000 inhabitants per year [2]. 

Prognosis of FL was significantly improved by the introduction of rituximab, a monoclonal anti-CD20-antiboby, in addition to chemotherapy. However, recurrences are frequent and the overall 10-year survival rate remains in the order of 80% [3]. Recognizing at an early stage which patients belong to these prognostically unfavorable forms is crucial in order to improve results. Prognostic indices such as the Follicular Lymphoma International Prognostic Index (FLIPI), the FLIPI-2, the PRIMA-PI and the m7-FLIPI, which incorporates specific mutations, all aim to classify patients into defined risk groups. However, evidence for the selective use of certain treatment strategies in FL [4,5] according to these scores is still scarse. Furthermore, according to a current consensus, the aforementioned indices are suitable for risk assessment in FL, but they are unsuitable for evaluating potential and effective therapies [6,7]. This conclusion leads to an ongoing need for early identification of risk factors in order to optimize early treatment decision in patients with FL.

The combined ^18^fluorodeoxyglucose-positron emission tomography/computed tomography (^18^F-FDG-PET/CT) is an established and widely used method in Hodgkin lymphoma (HL) and most FDG-avid NHL subtypes for both the detection of the tumor stage and response evaluation [8,9]. This method enables the evaluation of glucose metabolism semi-quantitatively by Standardized Uptake Value (SUV) and allows conclusions to be drawn about the presence of malignant lesions. The maximum Standardized Uptake Value (SUVmax) has been most frequently used in studies to determine the disease activity in lymphoma [10,11].

Numerous recent studies have shown that quantitative parameters like metabolic tumor volume (MTV) and total lesion glycolysis (TLG) are suitable parameters for validating the prognosis of B-cell lymphomas (Bcl) and HL [8,12,13]. In contrast, there have been very few studies evaluating the parameters MTV and TLG for the prognosis of FL after first-line therapy. In these studies, the “International Harmonization Project” (IHP) criteria and the “Deauville five-point Scale” (D-5PS) were primarily used to evaluate the success of therapy after initial treatment [14,15]. Currently, further studies focus on the importance of periodic reviews of tumor activity with ^18^F-FDG-PET/CT during and at the end of treatment aiming to improve prognosis of FL [16,17].

In the present study, the stage-related survival times of patients with localized or advanced follicular lymphoma after treatment, based on staging with ^18^F-FDG-PET/CT were examined. The main criteria for analyzing therapeutic outcomes were the assessment of progression-free survival (PFS), overall survival (OS) and lymphoma-specific survival (LSS) for 5- and 10-year periods. 

## 2. Materials and Methods

### 2.1. Patients 

The current retrospective study was reviewed and approved by the institutional ethics committee. The study included 48 patients who were treated in the Department of Radiation Oncology, University Hospital of Muenster, Germany from 2004 to 2016. The follow-up period ended on 30 November 2019. 

The following inclusion criteria were used: ● age ≥ 18 years, ● histologically confirmed FL, ● in all 48 patients an ^18^F-FDG-PET/CT was performed as a basic examination before the start of radiotherapy (RT). 

### 2.2. PET-CT Imaging and Interpretation

The ^18^F-FDG-PET/CT images of the patients included in the study were carried out on the PET/CT devices Biograph 16 and, since October 2013, Biograph mCT (Siemens, 91052 Erlangen, Germany). After a 4-h fasting period and a blood glucose level of less than 6.7 millimoles per liter (mmol/L), ^18^F-FDG was injected intravenously in a dose of 4 Megabecquerel (MBq) per kg (max. 350 MBq). Sixty ± 10 min after the injection imaging was acquired. In 23/48 patients, a non-contrast Low-Dose-CT from head to proximal high was acquired, with parameters 100–120 kilovolts (kV) and 15–25 milliampere-seconds (mAs). Three of these patients underwent an additional contrast-enhanced diagnostic CT scan. In 25/48 patients, a contrast-enhanced diagnostic CT-scan with variable technical parameters was performed. Following CT, PET emission scans were acquired at intervals of 2 min. Axial, coronal, and sagittal PET/CT Fusion images were reconstructed and SUV were calculated. SUV is defined as the ratio of the radioactivity concentration in a defined region to the whole body concentration of the injected radioactivity. Interfering factors preventing the determination of the SUV can be partial volume effects, high serum glucose levels and the time interval between the injection of ^18^F-FDG -PET/CT [18]. SUV max is the maximum uptake in a defined region. SUV peak is the average in uptake in a defined volume containing the maximum SUV. SUV mean is the average uptake in defined area. It is less affected by noise and therefore a more robust alternative. 

The PET/CT was evaluated according to the D-5PS (Table 1). First established in 2009 for interim-PET interpretation in HL and DLBCL [19], later also in FL [20], the D-5PS provides reproducible criteria for assessment of lymphoma response based on SUV measures (Table 1). One method is to compare SUVmax in residual tumor to SUVmax in liver and mediastinal blood pool [21]. To provide less susceptibility to volume effects and noise, it is recommended to compare SUV peak in residual tumor to SUV mean in the liver [22].

A retrospective evaluation of Deauville Scores in all PET/CT Scans was performed by one reader, on a visual basis or by comparing the maximum SUV of the tumor and the liver/mediastinum.

A second, semi-automatic reevaluation of all available scans (28/48) was performed by the same reader, using SyngoVIA (version VB30A-HF04; Siemens Healthineers) isocontour region-of-interest tool (default threshold, 40% of maximum) with semiautomatically calculated Deauville Scores based on the SUV peak in a spheric volume of 1 cm^3^ in the FDG maximum tumor site, the SUV mean in a spheric volume of 3 cm diameter (14.1 cm^3^) in the liver and SUV mean in a cuboid volume of 2 cm^3^ (2 × 2 × 1 cm) in the thoracic aorta. As proposed by several studies, PET positivity was defined as Deauville Score (DS) 4 and 5 [23].

After the staging of the 48 patients, the groups were divided based on the Ann Arbor classification: group 1 with 24 patients belonging to stages I and II and group 2 with 24 patients belonging to stages III and IV. The number of patients were 12 at stage I, 12 at stage II, 11 at stage III and 13 at stage IV according to Ann Arbor.

### 2.3. Radiation Techniques and Volumes

The target volume definition and radiation planning was generated in most cases using PET-CT or CT based 3D-CRT (three dimensional-conformal radiotherapy) or IMRT (intensity-modulated radiotherapy). 3D-CRT and IMRT allow an exact adaptation of the dose distribution to the target volume and to the organs at risk. Two dimensional based radiotherapy (2D-RT) was applied in only one patient.

Radiation treatment was administered via Extended-Fields, Involved-Field and Involved-Site target volumes. In Extended-Fields, the initial lymphoma manifestation, the pathological lymph nodes as well as adjuvant non-involved lymph nodes are radiated. The following Extended-Fields were used: Total Body Irradiation (TBI), Total Lymphatic Irradiation (TLI), Total Abdominal Irradiation (TAI), Mantle-Field radiation and Cranio-Spinal Irradiation (CSI). In Total Body Irradiation, the entire body is treated but the lungs are individually protected against radiation doses > 8 Gy. In contrast to Extended-Fields, the Involved-Field only covers the initial lymphoma manifestation and the pathologically affected lymph nodes with a safety margin of 1–2 cm. Involved-Site radiotherapy is also limited to the lymphoma manifestation and the nodes affected by lymphoma with a smaller safety margin of about 1 cm.

### 2.4. Data

In addition to epidemiological data such as gender and age, clinical pathological information of the patients (Ann Arbor stages, B symptoms, lactate dehydrogenase (LDH) level, blood count, FLIPI-index, DS, as well as expression of CD 20, Bcl-2 and Bcl-6), histological findings of tumor biopsies and bone marrow punctures were analyzed from the patient’s medical record. 

RT data on the type and technique, the total dose and fractionation, and the initial responses to radiation were collected and recorded. 

Additive chemotherapy treatments were noted and registered, as well as the total dose and the number of chemotherapeutic cycles. Relapses after RT and Radio-Chemotherapy (RChT) were analyzed to determine whether they were located inside or outside the radiation field of the first treatment and whether a histological transformation occurred. Furthermore, the therapy method used for the treatment of relapses and patient follow-up exams and tests were analyzed. Furthermore, the Common Toxicity Criteria (grade 1—mild; 2—moderate; 3—severe; 4—life threatening) were registered during the follow-up [24]. 

### 2.5. Statistical Analysis 

All statistical analysis was accomplished by using the SPSS software from IBM SPSS Statistics Version 25. Survival times were defined as follows:

● Progression-free survival time (PFS) was defined as the time interval between the beginning of RT and progression (increase in uptake of ^18^F-FDG-PET/CT, tumor growth, suspicious clinical symptoms and laboratory findings), death or last follow-up.

● Overall survival time (OS) was defined as the time interval between beginning of RT and death or the last follow-up.

● Lymphoma-specific survival time (LSS) was defined as the time interval between the beginning of RT and the lymphoma-relevant cause of death or the last follow-up.

Diagrams for survival times were done by using R and graphically represented by Kaplan Meier curves. Differences between the patient groups were determined using the log-rank test. 

Furthermore, univariate and multivariate hazard ratios (HRs) ware calculated using Cox regression analysis with corresponding 95% confidence intervals (CIs).

## 3. Results

### 3.1. Patient Characteristics

The median age of the 24 patients with tumor stages I/II was 52 years (range 18–76) with a female to male ratio of 10 to 14. The median age in the stage III/IV group was 58.5 years (range 43–85) and of the 24 patients in this group, 13 were female and 11 were male (Table 2). In ECOG performance status, 22 stage I/II patients and 22 stage III/IV patients had a score of 0 to 1. Twenty-two patients in stages I/II and 22 patients in stages III/IV had FL of Grade 1/2. Using the FLIPI index, 18 stage I/II patients had a low risk and 6 had an intermediate risk FLIPI, whereas in the stage III/IV group 17 patients had a high risk and 7 patients had an intermediate risk FLIPI, respectively. Out of 24 patients with stages I/II, 22 patients had a positive expression of CD-20, 12 patients expressed Bcl-2 and 10 patients Bcl-6. Twenty stage III/IV patients showed positive expression of CD-20, 14 of Bcl-2 and 2 of Bcl-6 (Table 2). 

### 3.2. Staging

The classification using the DS before RT showed 9 patients (37.5%) with a scoring of 5 in stages I/II and 15 patients (62.7%) with a score of 5 in stages III/IV (Table 3).

The semi-automatic DSs could be experimentally redetermined for 28 of the 48 PET/CTs. Because the model could only be applied to PET/CTs after 2007, 20 scores could not be determined retrospectively. For 26 PET/CTs, both methods of determination resulted in the same DSs (Table 4). Two cases showed a minor deviation between the visual and semi-automatic DS, with the semi-automatic score being one level higher than the visually resp. manually calculated score, the results of DS calculation in one of these patients are illustrated in Figure 1.

All patients underwent an ^18^F-FDG-PET/CT before radiation treatment to determine the target volume. Sixteen patients in stages III and IV and nine patients in stage II received a systemic therapy before radiation. In 38 of 48 patients, CT and PET/CT showed the same lymphoma manifestations (79.2%) (Table 5). These 38 patients include three patients in stage I who got a lymphoma extirpation before radiation. Therefore, PET and CT did not show any enlarged lymph nodes or increased metabolism anymore. For another three cases of patients in stage III and for one patient in stage IV, the PET and CT have shown the same stage but there was a difference between the number of enlarged lymph nodes and the lower number of lymph nodes with an ^18^F-FDG-uptake: in these four cases, some enlarged lymph nodes did not show an ^18^F-FDG-uptake after systemic therapy anymore (8.3%). In another stage I patient with an intestinal FL proven by biopsy, the CT did not show any abnormalities, whereas the PET showed an uptake in the intestine and this led to an upstaging in this case (2.1%). In another one patient stage I, two patients stage II, and one patient stage IV, the CT showed enlarged lymph nodes being suspicious for malignancy. In these cases, the PET did not indicate a metabolic-uptake and resulted in a downstaging in 4 patients (8.3%).

### 3.3. Treatment

Based on the PET-CT staging and the clinical condition of the patient, the treatment concept was determined in an interdisciplinary tumor conference.

Of 24 FL patients in stages I and II, 10 patients (41.7%) were treated with radiation as definitive therapy and 2 patients (8.3%) had lymphoma extirpation in the neck area before the radiation. Combined chemotherapy with the CD-20 antibody rituximab, either alone or in combination with cytostatics, cyclophosphamide, hydroxydaunorubicin, vincristine, and prednisone (R-CHOP) was performed in 4 patients (16.7%) before RT and a combined treatment approach with immunotherapy (rituximab), pre and post radiation, was applied in 8 patients. Three of these patients (12.5%) were included in the MabThera (rituximab) and Involved-Field radiotherapy study (MIR study). Intensity-modulated radiation therapy (IMRT) was performed in 10 stage I and stage II patients and 3D radiation was performed in 14 patients. Radiation doses between 30 Gy and 46 Gy were applied in fractionations from 1.5 to 2.0 Gy single dose (Table 6). Including the patients of the MIR study, 15 patients have received an Involved-Field radiation (62.5%). One patient was treated with an Involved-Site target volume (4.2%). Extended-Field radiation was applied in 8 patients (33.3%). Out of them three received a Total Abdominal Irradiation (12.5%), two patients got a Total Lymphatic Irradiation (8.3%), in three patients a Mantle-Field radiation was applied (12.5%).

Eight of 24 patients with advanced stages III and IV of follicular lymphoma (33.3%) were treated with radiotherapy alone. One patient out of these 8 patients with sole RT died during radiation treatment due to progressive disease. Further 16 patients (66.7%) with advanced stages were initially started with chemotherapy. The regimen included immunotherapy with rituximab and the combination therapies R-CHOP, high dose rituximab-BCNU-carmustine-etoposide-cytarabine-melphalan (HD-R-BEAM), dexamethasone-high dose cytarabine-cisplatin (DHAP), rituximab-ifosfamide-carboplatin-etoposide (R-ICE), plus etoposide and bendamustine plus interferon. One patient with stage III and 4 patients with stage IV died shortly after radiation series and 1 patient in stage IV died during radiotherapy due to rapidly progressing disease. Radiation was performed using different techniques: IMRT (7 patients), 3D-CRT (16 patients) and 2D-RT (1 patient). Radiation doses between 4 Gy and 50.4 Gy were applied in fractionations of 1.8 or 2 Gy single dose (Table 6). TBI was carried out with a total dose of 4 or 12 Gy, single dose of 2 Gy and two fractions per day. Involved-Field radiation was applied in seventeen patients (70.3%). Seven patients received Extended-Field radiation volumes (29.2%). Out of them, in three patients a Total Body Irradiation was applied (12.5%), a Cranio-Spinal Irradiation was performed in one patient (4.2%), two patients received a Total Lymphatic Irradiation (8.3%), a Mantle-Field radiation was carried out in one patient (4.2%).

### 3.4. Survival

After treatment, 23 out of 24 stage I and II patients showed a CR (95.8%), and out of 24 stage III and IV patients, CR was demonstrated in 17 patients (70.8%).

Partial remission (PR) was observed in one patient at stage II and 6 patients in stages III/IV had a progressive course during therapy.

The imaging methods US, CT and ^18^F-FDG-PET/CT were used selectively to determine PFS, OS and LSS over the 5- and 10-year course.

The 5-year PFS in stage I/II patients was 90.0% (95% CI 77–100) and 44.3% (95% CI 27–71) in stage III/IV patients, respectively. In the analysis of the 10-year PFS, 78.1% (95% CI 62–100) was shown in the stage I/II group and 28.5% (95% CI 13–63) in stage III/IV (Figure 2).

Stage I/II patients had a 5-year OS of 100% and in the stage III/IV groups, the OS was 53.7% after 5 years (95% C.I. 36–79). Ten-year OS was 93.3% (95% CI 80–100) in patients with stages I/II and 48.4% (95% CI 31–79) for stage III/IV patients (Figure 3). 

Regarding the 5-year LSS, the proportion of patients with stage I/II was 100% and in the group with stage III/IV 69.2% (95% CI 52–90), after an observation period of 10 years, the LSS was 93.8% (95% CI 81–100) in patients with stage I/II and 62.3% (95% CI 44–88) in stage III/IV (Figure 4).

Furthermore, we investigated how progression-free survival depends on PET-positivity in pre-RT PET/CT. Deauville score 4–5 as PET-positive scores were compared to Deauville score 1–3. Visual/manual DS curves differed with a p-value of 0.315. 5-year PFS for DS 1–3 was 72.0% and for DS 4–5 69.5%. 10-year PFS for DS 1–3 was 72.0% and 47.5% for DS 4–5 (Figure 5, Table 7). Semi-automatic DS curves differed with a p-value of 0.233. 5-year PFS for DS 1–3 was 79.0% and for DS 4-5 57.5%. 10-year PFS for DS 1–3 was 59.0% and 43.0% for DS 4–5 (Figure 6, Table 8).

### 3.5. Relapses

After a median follow-up period of 87.6 months (range of 1–186 months), 4 patients with stage I/II (16.7%) and 8 patients with stage III/IV (33.3%) revealed a relapse. The localization of the early-stage relapses varied. Two (8.3%) of the relapses were located within and 2 (8.3%) outside the irradiated region. Of the 8 late-stage relapses, 5 were localized within and 3 outside the irradiated area. Two patients with early stage (8.3%) and 2 patients with late stage FL (8.3%) developed secondary relapses. All 12 patients with recurrence received salvage therapy using combinations of radiation and antibody or high-dose chemotherapy. In 3 of the 4 patients with stages I/II FL, salvage therapy led to complete remission, and in the patient group with stages III/IV a complete remission was achieved in 7 of 8 recurrences. One patient had a stable and progression-free disease phase.

### 3.6. Toxic Side Effects

Acute toxic side effects were noted and were assessed by means of Common Toxicity Criteria [24]. Twenty-three with stages I/II and 17 patients with stages III/IV showed adverse reactions ranging from grade 1 to 3 (Table 9). Sixteen patients (67%) with stage I/II and 9 patients (37.5) with stage III/IV had mild side effects of grade 1 to 2. Twelve moderate side effects (grade 3) occurred in 7 patients (29%) with stages I/II and in 8 patients (33%) with stages III/IV. 

### 3.7. Univariate and Multivariate Analyses

Univariate and multivariate analyses were performed on baseline patient characteristics, including age, female sex, LDH, Bcl-2 and Bcl-6, no extranodal disease, FLIPI-Index, DS and Stage disease.

On univariate analysis, patients with absence of extranodal disease had a low risk of progression (HR, 0.283; 95% CI 0.112–0.719; *p* = 0.008 (Table 10). Furthermore, univariate analysis showed that stage III/IV diseases were significantly associated with higher risk of progression in comparison to stages I/II (HR, 0.194; 95% CI, 0.069–0.547; *p* = 0.002). Other parameters associated with a higher risk of progression were elevated LDH (HR, 1.007; 95% CI 1.003–1.010; *p* = 0.000), and high FLIPI-Index (HR, 2.428; 95% CI 1.365–4.318; *p* = 0.003). In a multivariate model, the only other factors associated with risk of relapse were high FLIPI-Index (HR, 2.422; 95% CI 1.302–4.513; *p* = 0.005) and elevated LDH (HR, 1.007; 95% CI 1.003–1.010; *p* = 0.000). 

## 4. Discussion

The present study analyzed the importance of ^18^F-FDG-PET/CT for stage-appropriate therapy in 48 patients with follicular lymphoma. Based on the diagnosis using ^18^F-FDG-PET/CT, 12 patients with stage I and 12 patients with stage II received stage-related therapy with predominantly radiation alone in stage I and chemoradiotherapy in stages II, III, and IV. The 5-year rates of OS and LSS stages I and II were 100% and the 5-year rate of PFS was 90% for both stages, which was significantly more than in stages III/IV.

While Zhou et al. have shown that Interim PET results based on D-5PS or IHP criteria were not significantly correlated with PFS [11], our methodological approach aims at evaluating the prognostic value of the DS of the pre-RT ^18^F-FDG-PET/CT. PET-positivity was defined as DS ≥ 4. To avoid false-positive and false-negative interpretations, a lower cut-off (score ≥ 3) and a higher cut-off (markedly above the liver, score ≥ 5) were not used. Deauville Score 4 and 5 were correlated with shorter PFS compared with DS 1–3, especially in a smaller subgroup of patients with semi-automatically assessed DS, but the difference did not reach statistical significance, given the relatively small number of semi-automatically evaluated patients and the different therapy regimes before PET/CT. 

The importance of ^18^F-FDG-PET/CT for selective, stage-appropriate therapy has furthermore been highlighted by several recent studies, whereby it is relevant that follicular lymphoma is mostly diagnosed in advanced stages III and IV. A recent research review in 2012 by Jacobson et al. focuses on early diagnosis, especially as local radiation therapy is becoming increasingly important for patients with early stages, which accounts for 20–25% of the FL population [25]. Ahmed et al. in 2013 concluded that among stages I and II patients, definitive local control and freedom from recurrence for up to 10 years can be achieved with radiotherapy alone [26]. A study from Filippi et al. in 2016 suggested optimizing diagnostics by using ^18^F-FDG-PET/CT and optimizing treatment by using multiple approaches in addition to radiotherapy such as CHOP and anti-CD20 antibody therapy [27]. This study raises the problem of imprecise differentiation of stages I and II of the FL. A study by Ng et al. from 2019 focused on the lack of differentiation [28]. The authors compared staging methods for 47 FL patients using CT and ^18^F-FDG-PET/CT. A significantly longer PFS was shown in the group of patients who were assigned to stage I by means of diagnostics using ^18^F-FDG-PET/CT. The researchers recommended this diagnostic differentiation of stages I and II as an essential prerequisite for RT alone in a stage I FL.

There are also critical comments with regard to this categorical recommendation. In a systematic review of the value of ^18^F-FDG-PET/CT for the staging of FL, Adams et al., 2017 point out that patients are upstaged by FDG-PET compared to CT, and that the current studies on this topic show methodological errors [29]. There is a lack of data on FDG-PET-induced FLIPI risk stratification changes compared to CT [29]. The authors further argue that well-designed studies are required before FDG-PET can be recommended for routine staging of follicular lymphoma. The results of a multicenter study by the International Lymphoma Radiation Oncology Group (ILROG) contradict this assessment. It was shown that the result after RT in patients staged according to PET-CT was better than in previous series, especially in stage I disease. Therefore, it could be concluded that the results of the RT for actually localized FL were underestimated [30]. In comparison to ILROG who described an upstaging in 10–60% of the patients, our study as based on a limited number of cases shows an upstaging in 2.1% and a downstaging in 8.3%. In a further study by the ILROG, it was investigated that approximately 30% of patients relapse within 5 years, although they had been classified as stage I by ^18^F-FDG-PET/CT [31]. It could be shown that for relapsing patients, previous RT was associated with a better prognosis. In more recent studies, the combination of nodal FL irradiation and the anti-CD20 antibody rituximab (MIR study) was examined in stages I and II [32]. The primary endpoint of the study was progression-free survival (PFS) after 2 years. PFS and OS were 78% and 96% after 5 years, with a mean follow-up of 66 and 78 months, respectively. The MIR study was able to show that Involved-Field radiation therapy in combination with rituximab is well tolerated, is highly efficient and has low recurrence rates in the early stages of FL. Compared to other therapeutic approaches with higher aggressiveness, the MIR study shows comparable effectiveness of early treatment, more than five years of survival, but without impairment in the quality of life that comes with aggressive approaches. In a follow-up study with 107 patients who received combined radioimmunotherapy (RIT), the efficacy of therapy and recurrence patterns after RIT in early stages and extranodal FL were explored. It was shown that, comparable to the results of the MIR study, extranodal involvement and a grade 3A histology do not have a negative impact on PFS [33]. 

In the present study univariate analyses have shown, that stages I or II and the absence of extranodal disease were significantly associated with lower risk of progression independent of each other. In the multivariate Hazard-model, the only factors associated with risk of relapse were FLIPI-Index and LDH (Table 10). In accordance with the presented study, the ILROG-study [30] had consistent univariable results regarding stage II disease, age and sex, but there were different results regarding FLIPI-Score and Bcl-2 and Bcl-6 expression. 

In contrast to the present study, in the ILROG-study, the FLIPI-scores had no significant influence regarding the PFS [30]. The FLIPI-score enables the evaluation of tumor progression and remains valid even if immunochemotherapy is used [4]. 

Regarding Bcl-2 expression, the ILROG-study showed a significant association with higher risk of progression in stage II disease on univariable analysis (HR, 2.34; 95% CI, 1.66–3.30; *p* = 0.0001). Bcl-2 overexpression is present in 80% to 90% of patients with FL, but there were no significant influences on PFS in the present study or in the ILROG-study [30]. We agree with the authors deduction, that at present it is not possible to draw any firm conclusion regarding Bcl-2 and prognosis of early-stage disease treated with RT, but the relationship should be further explored.

## 5. Conclusions

The present study confirms Involved-Field radiation in combination with immunotherapy as an optimal therapeutic approach for the treatment of early stage nodal and extranodal FL in accordance with the literature. ^18^F-FDG-PET/CT is a valid tool for staging, especially in early-stage FL in order to identify patients with excellent prognosis by RT as shown by OS rates and LSS rates over 90% at ten years. Furthermore, a lower DS in the pre-RT PET/CT correlates with longer PFS. 

## Figures and Tables

**Figure 1 biomedicines-09-00561-f001:**
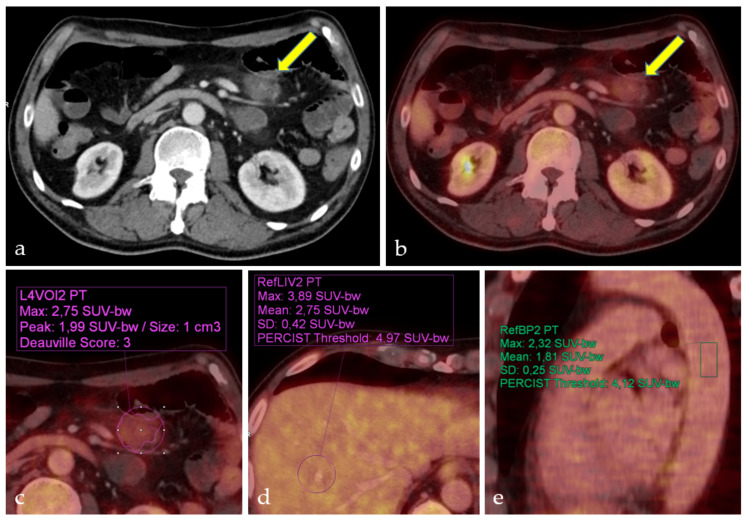
Deauville Score determination of mesenteric follicular lymphoma in stage IV. Patient with mesenteric follicular lymphoma (**a**) CT, (**b**) PET/CT (arrows). Visually assessed Deauville Score 2 (**b**). Semiautomatically calculated Deauville Score 3 (**c**), metabolic activity between liver (**d**) and mediastinal blood pool (**e**).

**Figure 2 biomedicines-09-00561-f002:**
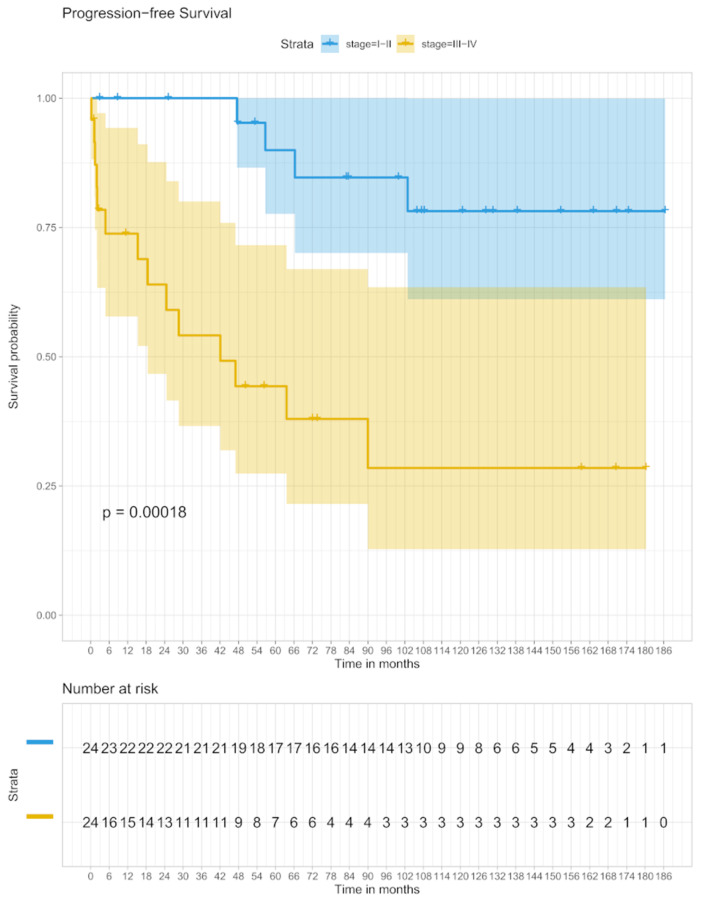
PFS in stage I/II and stage III/IV.

**Figure 3 biomedicines-09-00561-f003:**
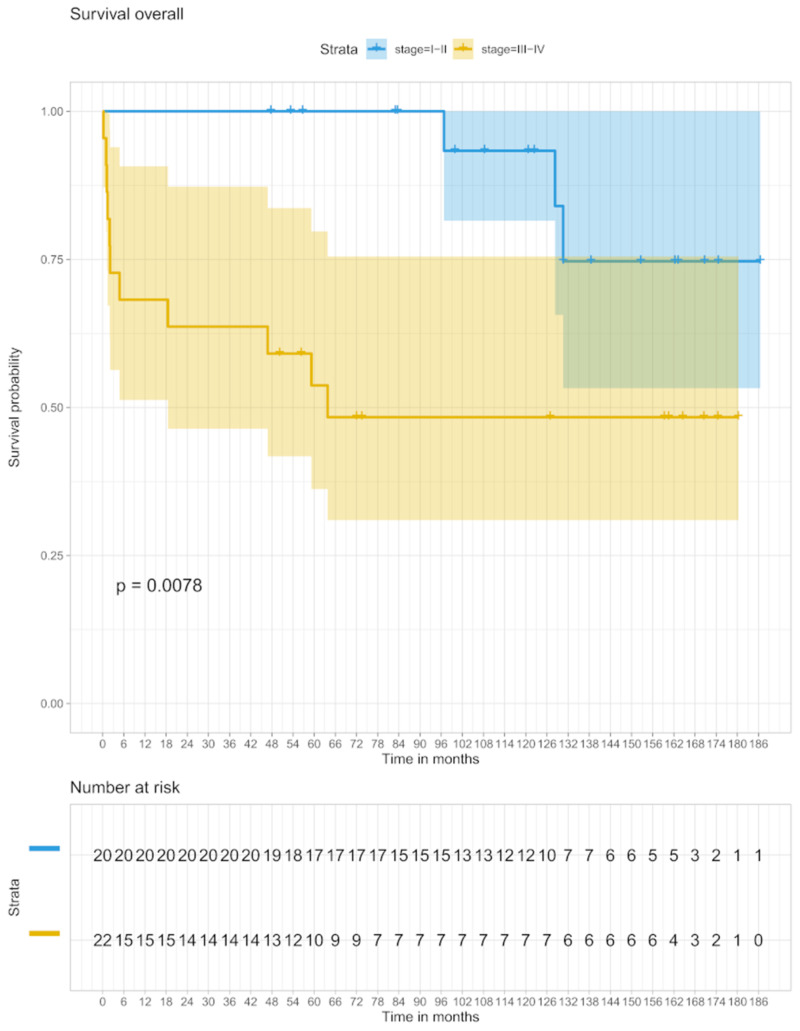
OS in stage I/II and stage III/IV.

**Figure 4 biomedicines-09-00561-f004:**
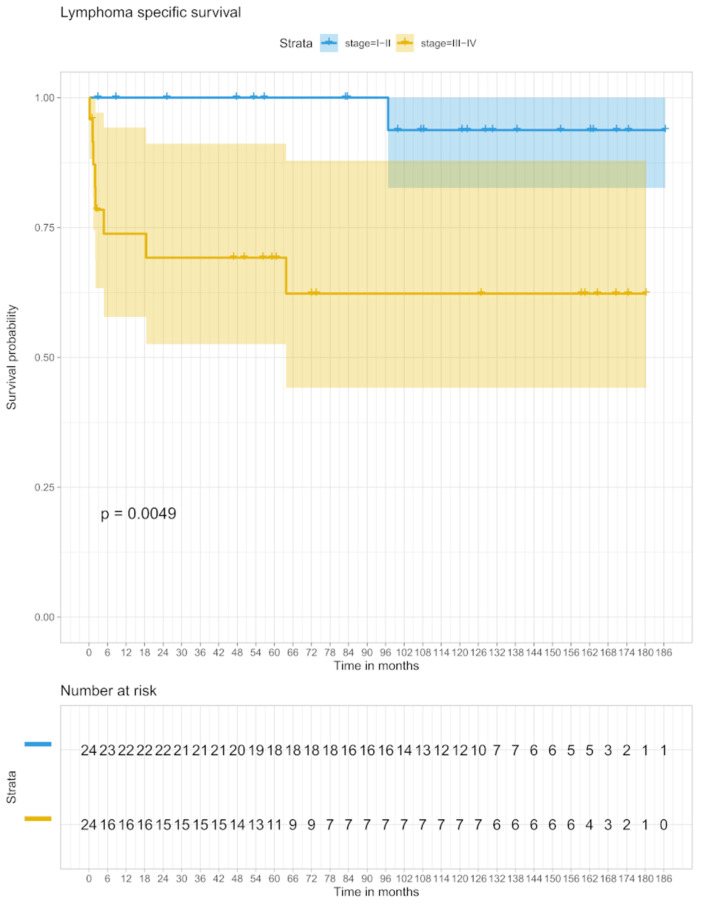
LSS in stage I/II and stage III/IV.

**Figure 5 biomedicines-09-00561-f005:**
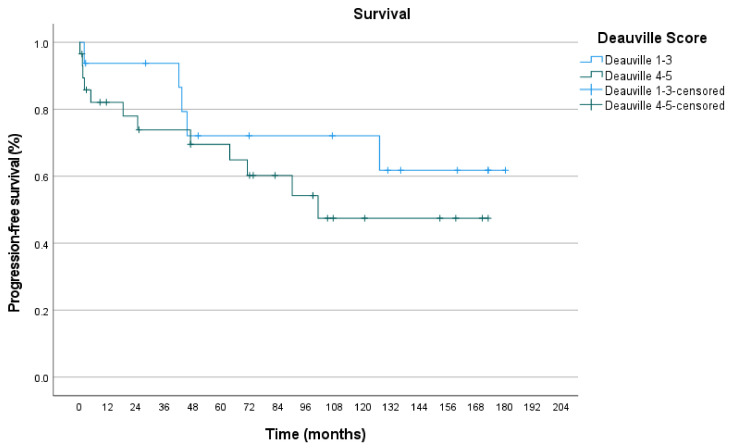
PFS for visual/manual DS.

**Figure 6 biomedicines-09-00561-f006:**
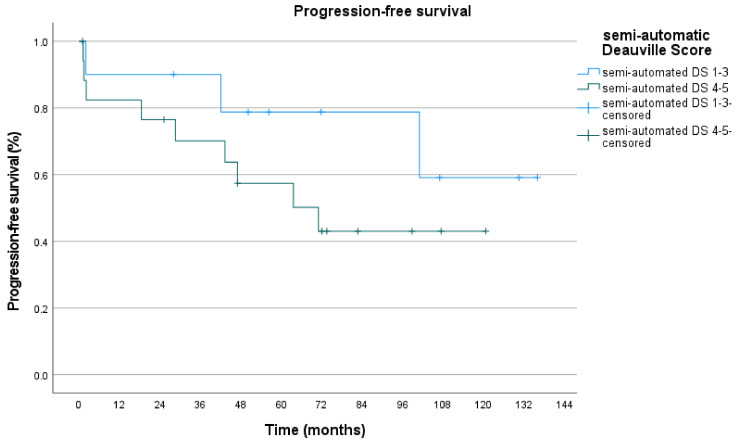
PFS for semi-automatic DS.

**Table 1 biomedicines-09-00561-t001:** Deauville five-point scale.

Score	Definition
1	No uptake
2	Uptake ≤ mediastinum
3	Uptake > mediastinum but ≤ liver
4	Uptake moderately increased above liver at any site
5	Uptake markedly increased compared to the liver at any site and/or new sites of disease

**Table 2 biomedicines-09-00561-t002:** Patients age, gender and lymphoma characteristics.

Parameter	Classification	Stage I/II n = 24 [100%]	Stage III/IV n = 24 [100%]
Median age	(range)	52 (18–76)	58.5 (43–85)
Sex	Female Male	10 (41.7) 14 (58.3)	13 (54.2) 11(45.8)
ECOG	0 1 2 Unknown	18 (75.0) 4 (16.6) 1 (4.2) 1 (4.2)	8 (33.3) 14 (58.3) 2 (8.3) 0
Stage	I II III IV	12 (50.0) 12 (50.0) 0 0	0 0 11 (45.8) 13 (54.2)
Grade	1 2 3A unknown	11 (45.8) 11 (45.8) 2 (8.3) 0	15 (62.5) 7 (29.2) 0 2 (8.4)
FLIPI	Low risk Intermediate risk High risk	18 (75.0) 6 (25.0) 0	0 7 (29.2) 17 (70.8)
CD-20-expression	Positive Negative Unknown	22 (91.6) 0 2 (8.3)	20 (83.3) 1 (4.2) 3 (12.5)
Bcl-2-expression	Positive Negative Unknown	12 (50.0) 10 (41.6) 2 (8.3)	14 (58.3) 4 (16.6) 6 (25.0)
Bcl-6-expression	Positive Negative Unknown	10 (41.6) 12 (50.0) 2 (8.3)	2 (8.3) 16 (66.7) 6 (25.0)
Chemotherapy	Systemic therapy No systemic therapy Rituximab	9 (37.5) 15 (62.5) 12 (50.0)	16 (66.7) 8 (33.3) 8 (33.3)
Radiation	Min. Max. Mean	30 Gy 46 Gy 36.4 Gy	4 Gy 50.4 Gy 32.4 Gy
FL manifestations	Only nodal Extranodal	20 (83.3) 4 (16.6)	10 (41.6) 14 (58.3)

**Table 3 biomedicines-09-00561-t003:** Visual/manual Deauville Scoring before radiotherapy.

Deauville-Score	Stage I/II n = 24 (100%)	Stage III/IV n = 24 (100%)
1	3 [12.5%]	2 [8.3%]
2	5 [20.8%]	2 [8.3%]
3	3 [12.5%]	1 [4.1%]
4	4 [16.6%]	4 [16.6%]
5	9 [37.5%]	15 [62.7%]

**Table 4 biomedicines-09-00561-t004:** Visual/manual versus semi-automatic DS.

Deauville Score	Stage I/II n = 14	Scores	Stage III/IV n = 14	Scores
1	1	V/M *: 1 S-A **: 3	2	V/M: 1 S/A: 1
2	2	V/M: 2 S-A: 2	1 1	V/M: 2 S-A: 2 V/M: 2 S-A: 3
3	3	V/M: 3 S-A: 3	0	-
4	1	V/M: 4 S-A: 4	4 1	V/M: 4 S-A: 4 V/M: 4 S-A: 5
5	7	V/M:5 S-A: 5	5	V/M: 5 S-A: 5

* V/M = visual/manual; ** S-A = semi-automatic.

**Table 5 biomedicines-09-00561-t005:** Staging before radiotherapy.

Stage	Staging before Radiotherapy
(n = 48)	[Stage]
I (n = 12)	
7	CT: I
	PET: I
3	CT: 0
	PET: 0
1	CT: I
	PET: 0
1	CT: 0
	PET: I
II (n = 12)	
10	CT: II
	PET: II
2	CT: III
	PET: II
III (n = 11)	
9	CT: III
	PET: III
2	
	CT: III *
	PET: III
IV (n = 13)	
6	CT: IV
	PET: IV
1	CT: IV*
	PET: IV
2	CT: III
	PET: III
1	CT: III *
	PET: III
2	
	CT: II
	PET: II
1	
	CT: III
	PET: I

In five patients, the PET influenced the classification of stage (n = 1 upstaging, n = 4 downstaging); * these four cases showed a lower number of ^18^F-FDG -positive lymph nodes, but without influence on the classification of stage.

**Table 6 biomedicines-09-00561-t006:** Radiotherapy modalities.

	Stage I/II	Stage III/IV
	n = 24 [100%]	n = 24 [100%]
Radiation volume		
Extended-Field	8 [33.3]	7 [29.2]
Involved-Field	15 [62.5]	17 [70.8]
Involved-Site	1 [4.2]	0
Radiation planning technique		
IMRT	10 [41.7]	7 [29.2]
3D-CRT *	14 [58.3]	16 [66.6]
2D-RT **	0	1 [4.2]
Radiation total doses		
4–16 *** Gy	0	4 [16.7]
30–35 Gy	13 [54.2]	5 [20.8]
36–40 Gy	8 [33.3]	12 [50.0]
41–45 Gy	2 [8.3]	2 [8.3]
46–50 Gy	1 [4.2]	1 [4.2]
Fractionation single doses		
2.0 Gy	12 [50.0]	17 [70.8]
1.8 Gy	9 [37.5]	7 [29.2]
1.6 Gy	2 [8.3]	0
1.5 Gy	1 [4.2]	0

* 3D-CRT = three dimensional conformal radiation therapy; ** 2D-RT = two dimensional radiation therapy; *** n = 3 TBI (Total Body Irradiation) patients: n = 2 with 4 Gy, n = 1 with 12 Gy. n = 1 patient who died during radiation series, therefore only received 16 Gy.

**Table 7 biomedicines-09-00561-t007:** Number at risk.

Time (Months)	0	12	24	36	48	60	72	84	96	108	120	132	144	156	168	180	192
Deauville Score 1–3	17	14	14	13	10	9	8	8	8	7	7	5	4	4	3	1	0
Deauville Score 4–5	31	22	21	19	17	17	15	12	11	7	7	7	7	5	2	0	0

**Table 8 biomedicines-09-00561-t008:** Number at risk.

Time (Months)	0	12	24	36	48	60	72	84	96	108	120	132	144
Deauville Score 1–3	10	9	8	8	7	5	4	4	4	2	2	1	0
Deauville Score 4–5	13	18	13	11	7	7	6	2	2	1	1	0	0

**Table 9 biomedicines-09-00561-t009:** Side effects in 48 patients related to the treatment of follicular lymphoma, regarding to the Common Toxicity Criteria.

Side Effects n = 48 pts.	Grade 0	Grade 1/2	Grade 3	Grade 4/5
Dysphagia	n = 32	n = 14	n = 2	0
Weight loss	n = 36	n = 12	0	0
Nausea/Vomitus	n = 33	n = 10	n = 5	0
Erythema	n = 39	n = 9	0	0
Xerostomia	n = 39	n = 9	0	0
Mucositis	n = 40	n = 5	n = 3	0
Diarrhea	n = 42	n = 4	n = 2	0

**Table 10 biomedicines-09-00561-t010:** Univariate and multivariate influences on PFS.

Variable	Univariate	Multivariate 1	Multivariate 2	Multivariate 3
**Age**				
HR	1.014			
95% CI	0.976–1.054			
*p*	0.470			
**Female Sex**				
HR	0.526			
95% CI	0.2–1.385			
*p*	0.193			
**LDH**				
HR	1.007		1.005	1.007
95% CI	1.003–1.010		1.001–1.009	1.003–1.010
*p*	0.000		0.007	0.000
**Bcl-2 negative**				
HR	0.63			
95% CI	0.195–2.044			
*p*	0.442			
**Bcl-6 negative**				
HR	0.883			
95% CI	0.276–2.831			
*p*	0.834			
**No Extranodal disease**				
HR	0.283		0.644	
95% CI	0.112–0.719		0.180–2.310	
*p*	0.008		0.500	
**FLIPI**				
HR	2.428	2.422		
95% CI	1.365–4.318	1.302–4.513		
*p*	0.003	0.005		
**Deauville Score**				
HR	1.18	0.974		1.103
95% CI	0.821–1.687	0.666–1.425		0.750–1.621
*p*	0.376	0.892		0.619
**Stage I/II**				
HR	0.194		0.248	
95% CI	0.069–0.547		0.045–1.367	
*p*	0.002		0.110	

## Data Availability

Data is contained within the article.

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
