# Peer review of "Radiotherapy in Follicular Lymphoma Staged by 18F-FDG-PET/CT: A German Monocenter Study"

_biomedicines, 2021, doi:10.3390/biomedicines9050561_

Round 1

Reviewer 1 Report

I think the study is quite interesting. However, the authors should better specify the impact of PET staging on clinical approach

Author Response

Dear reviewer,

thank you for your reviewing of this manuscript and your opinion on our paper.

Point 1: However, the authors should better specify the impact of PET staging on clinical approach 

Response 1: We understand the reviewer’s thoughts concerning the impact of PET staging on clinical approach. According to the reviewer’s comment we have added that PET-CT is the basis for an adequate staging of follicular lymphomas, which is used to determine the treatment concept in an interdisciplinary tumor conference. Moreover, PET-CT has got an important role in determining the target volume during radiation planning.  

The modified passages and the passages answering the reviewer’s comments are highlighted in the manuscript. We hope that our revised manuscript is now suitable for publication in Biomedicines.

Reviewer 2 Report

Radiotherapy (RT) methods need to be described more in detail.

No data on fractionation was provided.

No data on RT fields, except 3 patients with involved field RT (MIR study).

Please describe extent of clinical or planning target volumes used for these patients.

Lines 217-220 : these 8 patients received 3DCRT and also IMRT too ?, 3 times of IMRT ?, please explain more clearly.

I suggest to make a table just for describing RT methods in detail.  

Author Response

Dear reviewer,

thank you for your reviewing of this manuscript and your opinion on our paper.

Point 1: Radiotherapy (RT) methods need to be described more in detail.

No data on fractionation was provided.

No data on RT fields, except 3 patients with involved field RT (MIR study).

Please describe extent of clinical or planning target volumes used for these patients.

Lines 217-220 : these 8 patients received 3DCRT and also IMRT too ?, 3 times of IMRT ?, please explain more clearly.

I suggest to make a table just for describing RT methods in detail.

Response 1: We understand the reviewer’s thoughts concerning the radiotherapy methods. According to the reviewer’s comment we have added a table describing radiation fields, radiation-planning techniques, total doses and fractionations. To complete the method section, a definition of the radiation fields and techniques was added. The frequencies of the radiation techniques, fields, doses and fractionation are presented according to early and late stages and are also shown in the results section. By describing in detail the nuclear medical aspects of the Deauville Score and the radiation volumes and techniques, the scientific design of the paper is enhanced. The details on radiotherapy methods also highlight the conclusions of the manuscript.

Responses relating to general Reviewer 2 questions:

These issues were also addressed by Reviewer 2.

Question: Does the introduction provide sufficient background and include all relevant references?

Response: The relevant scientific publications cited in the introduction describe the significance of PET-CT for correct staging and evaluation of treatment response.The added part in the results section about the important role of PET-CT for the treatment concept is supported by the publications of Chang CC et al. and Trotman et al., as quoted in the introduction. A comprehensive correct staging ist he basis for an effective onkological therapy. On that basis the introduction and cited references provide sufficient and relevant background.

Question: Is the research design appropriate?

Response: We thank the reviewer for this comment and hereby note that the description of the study design now includes an addition of the radiotherapeutic aspects. In combination with the nuclear medicine aspects it provides a clearer picture of the treatment structures.

Question: Are the methods adequately described?

Response: We thank the reviewer for this aspect and and refer to the first response. In the method section are inserted additional explanations about the exact process of the radiation planning and the definitions of applied target volumes.

Question: Are the results clearly presented?

Response: The order in which radiation results are presented has been revised for the better understanding of the reader and the additional results have been inserted, as starting in line 237 of the manuscript.

Question: Are the conclusions supported by the results?

Response: We thank the reviewer for this question. On the basis of the added details on the radiotherapy methods (lines 120-140), the conclusions now are more consistent and better supported by the results on radiotherapeutic.

The modified passages and the passages answering the reviewer’s comments are highlighted in the manuscript. We hope that our revised manuscript is now suitable for publication in Biomedicines.

Round 2

Reviewer 2 Report

Not all extended-field RT methods need to be described. The others except for a TBI, that was used in this study, better be discarded. 

Author Response

Dear reviewer,

thank you for your re-reviewing of this manuscript and your opinion on our paper.

Point 1: Not all extended-field RT methods need to be described. The others except for a TBI, that was used in this study, better be discarded.

Response 1: We understand the reviewer's thoughts regarding the detailed, methodological description of all Extended-Fields. In accordance with the reviewer's comment, we have limited the method section to the most important statements in order to make it easier for the reader to understand.

The modified passages and the passages answering the reviewer’s comments are highlighted in the manuscript. We hope that our revised manuscript is now suitable for publication in Biomedicines.